# The Potential Roles of Dietary Anthocyanins in Inhibiting Vascular Endothelial Cell Senescence and Preventing Cardiovascular Diseases

**DOI:** 10.3390/nu14142836

**Published:** 2022-07-10

**Authors:** Yonghui Dong, Xue Wu, Lin Han, Ji Bian, Caian He, Emad El-Omar, Lan Gong, Min Wang

**Affiliations:** 1College of Food Science and Engineering, Northwest A&F University, Xianyang 712100, China; dongyonghui2020@163.com (Y.D.); wuxue000912@163.com (X.W.); hanlin730@163.com (L.H.); caian.he@nwafu.edu.cn (C.H.); 2Kolling Institute, Sydney Medical School, Royal North Shore Hospital, University of Sydney, St. Leonards, NSW 2065, Australia; jbia3972@uni.sydney.edu.au; 3Microbiome Research Centre, St George and Sutherland Clinical School, University of New South Wales, Sydney, NSW 2052, Australia; e.el-omar@unsw.edu.au

**Keywords:** anthocyanin, gut microbiota, vascular endothelial cells senescence, clearance of senescence cells, cardioprotection

## Abstract

Cardiovascular disease (CVD) is a group of diseases affecting the heart and blood vessels and is the leading cause of morbidity and mortality worldwide. Increasingly more evidence has shown that the senescence of vascular endothelial cells is the key to endothelial dysfunction and cardiovascular diseases. Anthocyanin is a type of water-soluble polyphenol pigment and secondary metabolite of plant-based food widely existing in fruits and vegetables. The gut microbiome is involved in the metabolism of anthocyanins and mediates the biological activities of anthocyanins and their metabolites, while anthocyanins also regulate the growth of specific bacteria in the microbiota and promote the proliferation of healthy anaerobic flora. Accumulating studies have shown that anthocyanins have antioxidant, anti-inflammatory, and anti-aging effects. Many animal and in vitro experiments have also proven that anthocyanins have protective effects on cardiovascular-disease-related dysfunction. However, the molecular mechanism of anthocyanin in eliminating aging endothelial cells and preventing cardiovascular diseases is very complex and is not fully understood. In this systematic review, we summarize the metabolism and activities of anthocyanins, as well as their effects on scavenging senescent cells and cardioprotection.

## 1. Introduction

Cardiovascular diseases are currently identified as a major cause of death worldwide and will continue to exist for many years, placing considerable burdens on world health resources [1]. Endothelial cells are monolayer epithelial cells lining inside the vessel that directly contact with the blood. They play an important role in maintaining vascular integrity and functional regulation by regulating blood flow and fibrinolysis, vascular tension, angiogenesis, monocyte/leukocyte adhesion, and platelet aggregation [2]. Normal vascular endothelium is the gatekeeper of cardiovascular health, and abnormal vascular endothelium is the main factor leading to cardiovascular diseases such as atherosclerosis, aging, hypertension, obesity, and diabetes. At present, increasingly more studies have confirmed that vascular endothelial cell senescence may play a key role in endothelial dysfunction and aging-related vascular diseases [3].

Anthocyanidin is a kind of water-soluble natural pigment that widely exists in natural plants [4]. Like other natural flavonoids, anthocyanin has a C6-C3-C6 carbon skeleton. Due to the different carbon substituents (-OH, -OCH3) on the B ring, different types of anthocyanins were derived. The six common anthocyanins were Pelargonidin (Pg), Cyanidin (Cy), Delphinidin (Dp), Peonidin (Pn), Petunidin (Pt), and Malvidin (Mv). In addition to giving food a variety of bright colors, anthocyanin also has important biological activities, such as antioxidant, anti-inflammatory, and anti-aging effects, among others. In the past two decades, a large number of studies have shown that dietary anthocyanins have a good preventive effect on cardiovascular diseases [5].

This review aims to evaluate and clarify the mechanism of dietary anthocyanin in the clearance of senile vascular endothelial cells and the prevention of cardiovascular diseases. Therefore, the first part of the article focuses on the structure and function of anthocyanin and the bioavailability and metabolism after anthocyanin intake were mentioned. The second part introduces cell senescence and its mechanism and the relationship between vascular endothelial cell senescence and cardiovascular disease. Finally, we summarized and discussed the recent literature on anthocyanin inhibiting endothelial cell senescence.

## 2. Anthocyanin

### 2.1. Structure of Anthocyanin

Anthocyanin is a water-soluble flavanol compound that widely exists in fruits, vegetables, and flowers, such as blueberry, sunflower, grape, pitaya, purple sweet potato, and purple cabbage. It is an extremely important secondary metabolite in plants. The structure of anthocyanin is mainly composed of C6-C3-C6 as the basic C skeleton. The differences between anthocyanin molecules are mainly due to the number of hydroxyl groups, the type and bonding position of sugars, and the type and bonding position of acyl groups of modified sugar molecules. There are six kinds of anthocyanins in plants. When positions 3, 5, and 7 of a and C rings are Oh, anthocyanins are aglycones, mainly including delphinidin (12%), cyanidin (50%), pelargonidin (12%), petunidin (7%), malvidin (7%), and peonidin (12%) (Figure 1 and Table 1). Delphinidin and its derivatives, petunidin and malvidin, are the sources of blue and purple, while cyanidin and pelargonidin are the main pigments of bright red fruits. Under natural conditions, the free anthocyanin is unstable, so it is rare that anthocyanin mainly exists in the form of glycoside. The hydroxyl at positions 3, 5, and 7 of anthocyanin can form anthocyanin through glycosidic bond with one or more monosaccharides (glucose, galactose, etc.), disaccharides (rutinose, etc.), or trisaccharides. Due to the different types, positions, and quantities of sugars that are glycosides of anthocyanin, the types of anthocyanin formed are also different. At present, there are more than 250 known natural anthocyanins.

### 2.2. Physiological Activities of Anthocyanins

#### 2.2.1. Anti-Cancer

Cancer is a disease caused by uncontrolled growth and progressive development of abnormal cells, killing millions of people every year. By 2030, there will be more than 20 million new cancer cases. Anthocyanins have been shown to have the ability to inhibit the initiation, promotion, and progression of various cancers, such as colon cancer [6], liver and bladder cancer [7], breast cancer [8], brain cancer [9], kidney cancer and skin cancer [10], gastric cancer [11], and thyroid cancer [12]. The ability of anthocyanins to inhibit tumorigenesis and development is closely related to their ability to enhance antioxidant defense; exert anti-inflammatory effects; and interfere with ERK, JNK, PI3K/Akt, MAPK, and NF-κB signaling pathways. Yun et al. reported that purple grape anthocyanins prevented tumor-necrosis-factor-α-induced NF-κB activation by inhibiting IκBα phosphorylation and resisted the invasion of human colon cancer cells in a dose-dependent manner [13]. Fragoso et al. proved through experiments that cyanidin-3-O-rutinoside at 25 µmol/L can effectively reduce the motility of human colon adenocarcinoma cells, reduce the metastasis of cancer cells, and play an anticancer effect [14]. Mazewski et al. reported that anthocyanins extracted from purple and red maize enhanced the expression of apoptotic factors BAX, Bcl-2, cytochrome C, and TRAILR2/D5 and inhibited vascular endothelial growth factor in HCT-116 and HT-29 human colorectal cancer cell (Tie-2, ANGPT2, and PLG) expression to achieve anti-cancer efficacy [15]. The results of Lage et al. showed that black sweet cherry anthocyanins can inhibit the growth of breast cancer cells and have no toxicity to normal MCF-10A breast cells. Anthocyanins work against cancer by reducing oxidative stress, regulation of Akt/mTOR, p38, and survivin, preventing cancer cell proliferation and promoting apoptosis. Anthocyanins can significantly downregulate the mRNA expression of invasive/metastatic biomarkers (Sp1, Sp4, VCAM-1), and anthocyanins from black sweet cherry can effectively prevent and treat cancer [8]. Su et al. reported that hibiscus calyx anthocyanin could inhibit the growth, metastasis, and angiogenesis of B16-F1 cells by triggering PI3K/Akt and Ras/MAPK signaling pathways and downregulating the expression of VEGF and MMP-2/-9, which could effectively prevent and treat melanoma cancer [16]. Sugata et al. found that purple sweet potato anthocyanin blocked all stages of cell cycle by acting on cell cycle regulators (such as p53, p21, p27, Cyclin D1, and Cyclin A), thereby inhibiting the proliferation of breast cancer, colon cancer, and gastric cancer cells in a concentration- and time-dependent manner [17].

#### 2.2.2. Anti-Inflammatory

Inflammation is usually regulated by the body to secrete inflammatory cytokines and mediators. Therefore, it is generally believed that the downregulation of factor secretion may contribute to the treatment of diseases such as inflammation [18]. Epidemiology and research have shown that anthocyanin has an anti-inflammatory effect and can improve a variety of inflammation-related diseases, such as colitis [19], periodontitis, pharyngitis, and postprandial inflammatory response. Anthocyanin can change the redox state of cells and affect redox-sensitive inflammatory mediators through Nrf2-ARE signal modulation [20]. Hou et al. showed that anthocyanin inhibited COX-2 by inhibiting C/EBP, AP-1, and NF-κB, thereby reducing the production of pro-inflammatory cytokines IL-1β, IL-6, IL-8, and TNF-α [21]. Min et al. found that cyanidin-3-glucoside showed an effective anti-inflammatory effect by regulating NF-κB and MAPK activity [22]. Studies have shown that fenugreein can inhibit HcPT degradation, p65 nuclear translocation, and JNK phosphorylation, showing an indigenous anti-inflammatory activity [23]. In general, B-ring o-dihydroxyphenyl anthocyanin, such as fayashinin and cyanidin, has strong anti-inflammatory activity, while geranium pigment, peony pigment, and kumquat pigment do not show the above activity without o-dihydroxy structure [24]. Aboonabi et al. showed that 320 mg anthocyanidin daily intake in people with metabolic syndrome can significantly inhibit the expression of NF-κB-pathway-related proinflammatory factor genes and enhance the expression of *PPAR-γ* gene to reduce the risk of inflammation [25]. Duarte et al. showed that geranium pigment-3-O-glucoside in strawberry could inhibit the activation of IkB-α and reduce the phosphorylation of JNK-MAPK, leading to the decrease in NF-κB and AP-1 activation factors in the inflammatory pathway stimulated by TLR4, indicating that geranium pigment-3-O-glucoside had an anti-inflammatory effect [26]. The study of Karnarathne and other studies have shown that anthocyanin from *Hibiscus* can inhibit the secretion of nitric oxide and prostaglandin E2 in LPS-induced endotoxic shock zebrafish, while down-regulating the expression of inducible nitric oxide synthase and cyclooxygenase 2. Furthermore, LPS inhibited the production of pro-inflammatory cytokines such as TNF-α, IL-6 and IL-12 in RAW 264.7 macrophages. Anthocyanin also inhibits LPS-induced TLR4 dimerization or cell surface formation, thereby reducing MyD88 growth and IRAK4 phosphorylation, thereby inhibiting NF-κB activity [27].

#### 2.2.3. Anti-Oxidation

Humans produce free radicals during metabolism. Excessive free radicals can lead to lipid, protein, DNA, RNA, and sugar oxidation, which is closely related to cancer, Alzheimer′s disease, Parkinson′s disease, autoimmune deficiency, diabetes, obesity, and other diseases. As a natural plant pigment, anthocyanin not only can be used as a colorant, but also has prominent antioxidant activity. Anthocyanins can scavenge reactive oxygen species (ROS) and reactive nitrogen (RNS), such as superoxide anion (O_2_^−^), singlet oxygen (^1^O_2_), peroxide free radical (RCOO^·^), hydrogen peroxide, hydroxyl free radical (OH^·^), and peroxynitrite anion (ONOO^−^) [28]. The phenolic ring, hydroxyl side chain, and double bond in the glycosylation reaction of anthocyanin are helpful to scavenge free radicals. Compared with cyanidins and philoxerin, anthocyanin lacking O-phenyl structure in the B ring (sunflower pigment, geranium pigment, petunia pigment, and peony pigment) had low DPPH radical scavenging efficiency. Peonidin has methyl at 3′ position and OH at 4′ position, which is more active than pelargonidin. As reported by Fukumoto and Mazza, the hydroxyl at the third position of the B ring enhances the activity. Similarly, delphinidin with hydroxyl at 3′, 4′, and 5′ is more effective than cyanidin with hydroxyl at only 3′ and 4′ [29]. Harakotr et al. reported that the anthocyanin extract of purple corn had strong DPPH radical scavenging activity, and the anthocyanin content in the extract was positively correlated with antioxidant capacity [30]. Matera et al. reported that cyanidins in radish buds could significantly inhibit the automatic oxidation of linoleic acid and scavenge hydrogen-peroxide-free radicals [31]. Coklar et al. reported that anthocyanin extracts from *Mahonia aquifolium* (cyanidins, delphinidin, malvidin, peonidin, pelargonidin) had strong DPPH and ABTS radical scavenging ability and FRAP reduction ability [32]. Lu et al. fed D-galactose-induced aging mice black rice anthocyanin extract (cyanidin-3-O-glucoside). The activities of superoxide dismutase and catalase in mice were significantly improved, and the content of malondialdehyde and the activity of monoamine oxidase were reduced. Black rice anthocyanin extract showed a strong anti-aging effect in mice [33]. Huang et al. studied the antioxidant effect of main anthocyanins in blueberry on endothelial cells. The results showed that brocade pigment and its two glycosides decreased the levels of reactive oxygen species (ROS) and xanthine oxidase-1 (XO-1), but increased superoxide dismutase (SOD) and heme oxygenase-1 (HO-1). Moreover, the presence of glycoside greatly improved the antioxidant capacity of malvidin [34].

#### 2.2.4. Protective Effect on the Liver

Daveri et al. fed high-fat diet mice with 40 mg anthocyanin/kg BW (cyanidins and delphinidins). The changes of chemokine MCP-1, cytokine TNF-α, macrophage marker F4/80, and enzyme NOS2 were measured. The results showed that anthocyanin played a role in preventing liver injury [35]. Jiang et al. showed that when carbon tetrachloride-induced liver injury mice were fed with cyanidin-3-O-glucoside 800 mg/kg BW, cyanidin-3-O-glucoside could significantly alleviate liver injury and prevent fibrosis in mice. Cyanidin-3-O-glucose can protect the liver by reducing liver oxidative stress, reducing liver cell apoptosis, inhibiting liver inflammatory response, and ultimately inhibiting the activation of liver star [36]. Arjinajarn et al. showed that the anthocyanin extract of riceberry bran could prevent gentamicin-induced liver injury in rats by inhibiting intracellular oxidative stress and the activation of NF-κB factor, reducing liver cell inflammation and apoptosis [37]. Zhang et al. found that purple sweet potato anthocyanin could effectively inhibit the production of reactive oxygen species in mice and inhibit the accumulation of liver fat induced by high-fat diet by activating adenosine-monophosphate-activated protein kinase (AMPK) signaling pathway [38]. Cai et al. studied the effects of different doses of purple sweet potato anthocyanin on the main liver function indexes, liver histological changes, and oxidation state of mice with alcoholic fatty liver, finding that medium dose of purple sweet potato anthocyanin had an obvious protective effect on the release of alanine aminotransferase (ALT) in the mice with liver injury [39].

#### 2.2.5. Lowering Blood Glucose 

Diabetes is a non-infectious endocrine metabolic disease that can lead to serious complications of various organs, and the number of patients with diabetes is increasing. Maintaining normal blood glucose level is a necessary condition for maintaining body function. In the human body, glucose homeostasis is controlled by various organs, including the pancreas, liver, and other tissues, as well as complex networks of hormones and neuropeptides. The pancreas plays a key role in glucose homeostasis by secreting hypoglycemic hormone insulin [40]. Purple corn anthocyanins have significant effects on β-cell function and insulin secretion, which can protect pancreatic β cells from high-glucose-induced oxidative stress and improve insulin secretion ability of β cells [41]. The liver is the main part of human body and plays a fundamental role in glycogen storage, plasma protein synthesis, and detoxification [42]. Studies have shown that anthocyanin-rich mulberry extract inhibits gluconeogenesis and stimulates glycogen synthesis by increasing AMPK phosphorylation in the liver [43].

#### 2.2.6. Anti-Aging

Oxidative stress is one of the main inducing factors of aging, and excessive expression of inflammatory factors, DNA damage, and a series of inflammatory reactions activated by NLPR3 and NF-κB can also promote the aging of the body [44]. Many studies have shown that anthocyanin has an anti-aging effect. Jin et al. fed aged mice with anthocyanin from purple sweet potato and found that compared with the control group, anthocyanin from purple sweet potato could significantly reduce the serum MDA level and improve the activities of SOD and GSH-PX, and low-dose anthocyanin could achieve the same effect as the equivalent amount of vitamin, indicating that anthocyanin from purple sweet potato could play a role in delaying aging by improving antioxidant activity [45]. Wang et al. showed that Cy-3-glu and Pg-3-glu treatments could significantly inhibit the galactosidase in the aging process of human retinal pigment epithelium (RPE) cells induced by visible light irradiation and play a protective role in anti-aging [46]. Gao et al. found that Ribes meyeri anthocyanins can promote the proliferation of neural stem cells, improve cell senescence phenotype, reduce ROS and senescence-associated *P16Ink4a* gene expression levels, increase DNA synthesis, and prolong telomeres [47]. Wei et al. showed that anthocyanin could maintain the stability of the redox system in plasma and liver structure, as well as reduce the levels of inflammatory factors such as IL-1, IL-6, and TNF-α in the liver. At the same time, the decrease in the expression levels of sensors (ATM and ATR), media (H2AX and γ-H2AX), and effectors (Chk1, Chk2, p53 and p-p53) in the DNA damage signaling pathway indicate that anthocyanin can slow down aging by inhibiting DNA damage [48].

#### 2.2.7. Other Effects

Qin et al. used purple sweet potato anthocyanin (PSPC 500 mg/kg/day) to orally take high-fat model mice. The results showed that PSPC corrected the abnormal metabolic indexes induced by HFD, including improving obesity, reducing fasting blood glucose concentration, and improving glucose tolerance [49]. Lee et al. found the effect of black soybean anthocyanin on obesity. The results showed that TC/HDLc/LDLc/HDLc of obese patients taking black soybean anthocyanin were significantly decreased [50]. Farrell et al. established a mouse model of hyperlipidemia and high-density lipoprotein dysfunction to explore and determine that an anthocyanin-rich blackcurrant extract (BEE) (13% anthocyanin) can prevent inflammation-related HDL functional damage and apolipoprotein E atherosclerosis. The results showed that the total cholesterol content in the aorta of mice was significantly decreased, and the aspartate aminotransferase (AST) and fasting blood glucose were decreased, indicating that blackberry may affect chronic inflammation-related HDL dysfunction by affecting liver gene expression [51]. In addition, studies have shown that purple sweet potato anthocyanin has a protective effect on the kidneys. Qun and other studies have found that purple sweet potato anthocyanin can significantly improve kidney injury in mice fed with high fat diet by reducing the production of AGEs and ROS and improving insulin sensitivity. Its protective effect is played by inhibiting the expression of TXNIP and RAGE and further inhibiting the activation of NLRP3 inflammasome and IKKb/NFκB pathway [44].

The physiological activities of anthocyanins are summarized in Table 2.

## 3. Bioavailability of Anthocyanins and Their Interaction with the Gut Microbiome

### 3.1. Bioavailability of Dietary Anthocyanins

Bioavailability is the fraction of the ingested dose that a compound enters into the systemic circulation and acts at a specific site [53]; the bioavailability of food-borne anthocyanin is the ratio of anthocyanin absorbed and utilized by an organism [54]. Bioavailability of anthocyanins in natural forms was once considered low at 1–2% [55]. However, the newly identified anthocyanin metabolites suggest that their bioavailability may be higher than previously proposed, and evidence from Czank et al. [56] suggests that the bioavailability of cyanidin-3-glucoside (C3G) is 12.38% when using anthocyanin isotope tracers. In fact, the bioavailability of anthocyanins is highly dependent on their chemical structures, such as their molecular size, glycosylation, and/or acylation patterns (acylation increases the stability of anthocyanins but reduces their absorption), degree of polymerization, conjugation, and/or combination with other compounds [57].

After the food rich in anthocyanin enters the mouth, anthocyanin can be released from the vacuole structure by chewing the food, and the food is mixed with saliva to hydrolyze anthocyanin. Enzymatic hydrolysis is caused by beta-glucosidase from saliva, oral epithelium, and oral microflora [58]. Afterwards, these compounds move in different regions of the gastrointestinal tract. In the stomach, lower pH values (1.5–4) provide favorable conditions for the stability of anthocyanins to persist in the form of glycoside. At the same time, the organic anion membrane carrier bilirubin ectopic enzyme in gastric mucosa can mediate anthocyanin transport [53]. Studies have found that the affinity of bilirubin ectopic enzyme to maternal anthocyanin is higher than its aglycone. It can be seen that bilirubin ectopic enzyme may be an important means of transferring anthocyanin into circulation to play an acute role [59]. The main part of anthocyanin absorption is the small intestine. Similar to other flavonoids, anthocyanins can be deglycosylation (glycosidic cleavage), producing lipophilic aglycones, and then passively transported into epithelial cells. Deglycosylation can be mediated by β-glucosidase in the intestine and lactose hydrolase at the brush border of intestinal epithelial cells. In addition, absorption may include active transport of intact glycoside to epithelial cells via sodium-dependent glucose transporter 1 (SGLT1) or GLUT2, and then deglycosylation via cytoplasmic β-glucosidase [60]. Meanwhile, anthocyanin degradation may also be the result of colonic microbiota activity. The unabsorbed anthocyanin enters the colon, and the anthocyanin arriving at the colon is exposed to 300–500 different bacteria, among which *Bifidobacterium*, *Pseudomonas*, *Proteus,* and *Clostridium* are the most abundant. Intestinal flora release many deglycosylation enzymes, cleave the glycosylation portion, produce aglycones, and further open the ring to produce different phenolic acids (such as protocatechuic acid (PCA) and vanillic acid, syringic acid, ferulic acid, and hippuric acid) or aldehydes [61]. Therefore, part of anthocyanin uptake along the gastrointestinal tract decreases, while part of phenolic acid increases. The anthocyanin degradation products may also be absorbed from the intestine through epithelial monocarboxylic acid transporters and further metabolized in the liver or kidney [62], thus entering the body cycle and being absorbed by target organs and tissues. If not absorbed, they will be discarded through urine and feces. At the same time, bile produced in the liver is collected in the gallbladder and secreted into the small intestine under the action of intestinal hormones caused by food intake. Bile secretion components can be reabsorbed through the hepatic-intestinal circulation (typical manifestation of bile acid) or discharged through feces and urine. Anthocyanins can be effectively circulated in bile by enterohepatic circulation, prolong the residence time in vivo, and contribute to phase II enzyme metabolism (Figure 2).

### 3.2. Interaction of Dietary Anthocyanins with the Gut Microbiome

The gut microbiota refers to the microbiota that contains tens of trillions of microorganisms living in the gut, and this microbiota consists of at least 1000 different species of identified bacteria [63]. Most of the bacteria in the intestine are divided into seven phyla, namely, *Firmicutes*, *Bacteroidetes*, *Proteobacteria*, *Clostridium*, *Verruciformes*, *Cyanobacteria,* and *Actinobacteria*, of which *Firmicutes* and *Bacteroidetes* account for more than 90% of human intestinal microbiota [64]. Some of the essential functions of the gut microbiota include vitamin production, regulation of lipid metabolism, production of short-chain fatty acids as fuel for epithelial cells, and regulation of gene expression [65], as well as an important functional role in the gut–brain axis relationship, having the ability to influence host nutrition and healthy enzyme activity and metabolism. Intestinal microflora are involved in anthocyanin metabolism to produce metabolites; anthocyanin and its metabolites are transferred throughout the body, playing antioxidant, anti-aging, and other biological activities [65]. On the other hand, anthocyanins and their metabolites regulate the growth of specific bacteria in microbial communities, promote the proliferation of healthy anaerobic bacteria, and inhibit pathogenic bacteria [64].

Early in vitro studies of anthocyanin metabolism by gut microbiota showed that bacterial metabolism involves the cleavage of glycosidic bonds and the breakdown of anthocyanin heterocycles [66]. The main human metabolites of Pg-3-O-glucoside, Cy-3-O-glucoside, Dp-3-O-glucoside, Pn-3-O-glucoside, and Mv-3-O-glucoside are 4-hydroxybenzoic acid, protocatechuic acid, gallic acid, vanillic acid, and syringic acid [67]. Protocatechuic acid is considered the main metabolite of the gut microbiota and can be further metabolized to ferulic acid and hippuric acid [68]. Other metabolites produced after partial degradation of anthocyanin include catechol, pyrogallol, resorcinol, tyrosol, 3-(3′-hydroxyphenyl) propionic acid, dihydrocaffeic acid, and 3-(4′-hydroxyphenyl) lactic acid [69]. In a study utilizing isotope-labeled cyanidin-3-glucoside, Czank et al. investigated the metabolism and pharmacokinetics of this anthocyanin in humans after ingestion of 500 mg. A thorough analysis of these metabolites showed that the B-ring-derived ferulic acid had the highest concentration in feces, followed by the A-ring-derived ferulic acid and protocatechuic acid [56]. Hanske et al. described the metabolism of Cy-3-O-glucoside in human-microbiota-associated rats, concluding that the enzymatic glycosylation of Cy-3O-glucoside could be catalyzed by intestinal bacteria, including *B. branchingis* and *C. capsularis*, and protocatechuic acid, 2,4,6-trihydroxybenzaldehyde, and 2,4,6-trihydroxybenzoic acid (gallic acid) were identified as the main colon metabolites [70].

At the colon level, anthocyanin was first metabolized by local microbial groups through deglycosylation, and then degraded to phenolic acids, mainly protocatechuic acid, vanillin acid, syringic acid, gallic acid, and p-coumaric acid [61]. The main bacterial groups that can metabolize anthocyanins are *Bifidobacterium* and *Lactobacillus*, which have probiotic effects. Zhu et al. confirmed that Cy-3-O-glucoside and melanin significantly increased the number of *Bifidobacterium* and *Lactobacillus* [71]. In addition, these bacteria have enzymes such as β-glucosidase, which are necessary to catalyze the release of glycoside from aglycones and provide the necessary energy for the reproduction of bacterial populations [72]. Anthocyanin intake also increases the abundance of *Bacteroides* and decreases *Firmicutes*. In the in vivo experimental model of high-fat diet, Wang et al. observed a decrease in the ratio of *Firmicutes* to *Bacteroides* in cecum contents [73,74]. At the same time, the regulation of anthocyanin on intestinal microflora will increase the bacteria producing short-chain fatty acids (SCFAs). SCFA, as a fuel that can provide energy for epithelial cells, can reduce intestinal pH and inhibit the proliferation of pathogens, thereby improving intestinal barrier and avoiding the translocation of pathogens and antigens [72].
Figure 2Scheme of anthocyanin absorption and metabolism [75]. Anthocyanins can interact with enzymes in the oral cavity after being ingested, resulting in enzymatic hydrolysis of anthocyanins. When reaching the gastric cavity, anthocyanins undergo a series of transformations under different digestive enzymes and acidic pH values and may be absorbed into the blood through active or paracellular transport. Anthocyanins reaching the intestine undergo phase I and II metabolism in the small intestine and are affected by intestinal microflora in the large intestine. All these newly formed metabolites and intact anthocyanins will be absorbed into the blood through a variety of mechanisms, and further metabolized in the liver or kidney, which enter the body cycle and are absorbed by the target organs and tissues. If not absorbed, they are discarded through urine and feces.
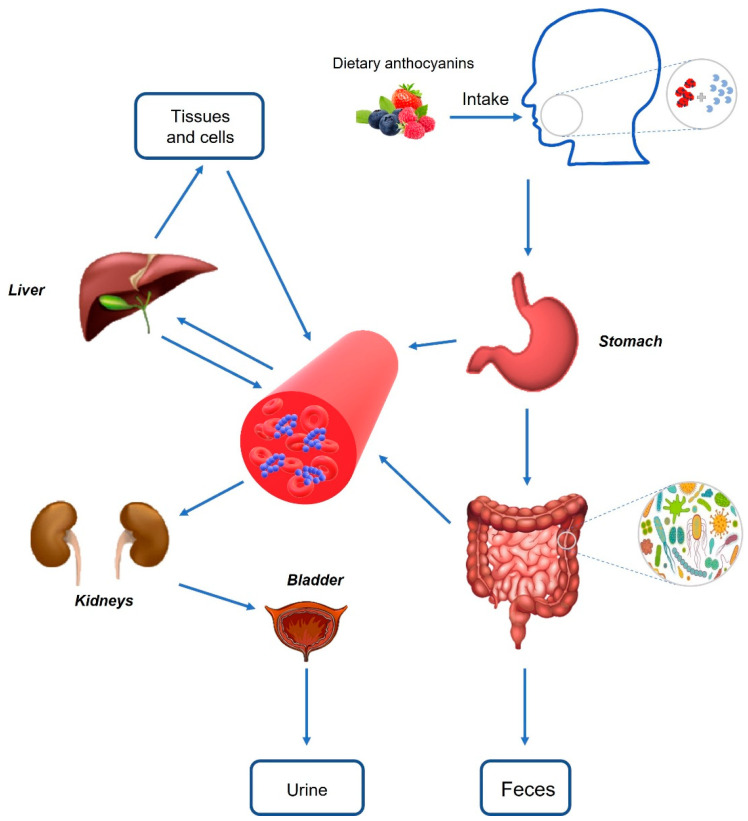


## 4. Effects of Anthocyanins on Endothelial Cell Aging and Cardiovascular Protection

### 4.1. Cell Senescence and Vascular Endothelial Cell Senescence

Cell senescence is a permanent cell cycle arrest with replication ability [76] accompanied by specific changes in cell morphology and gene expression and function (Figure 3). Cell senescence can be induced by many factors, and telomere shortening is one of the main signs of senescence. Due to incomplete lag chain synthesis during DNA replication, the number of telomere repeats decreases with each cell division. This loss is compensated by telomerase. In somatic cells without telomerase expression, as the number of cell division increases, telomeres gradually shorten to the limit, leading to DNA double strand breaks [77]. DNA double strand breaks induce DNA damage by activating ATM/ATR and Chk1/Chk2, thereby activating p53 signaling pathway and triggering cell senescence [78]. There is evidence that aging can be reversed by activating telomerase. In particular, when telomerase genes are reactivated in these aging mice, premature aging in telomerase-deficient mice can be reversed [79]. Excessive production of reactive oxygen species (ROS) can also challenge the integrity and stability of DNA, promote cell cycle arrest to induce premature senescence [80], or directly regulate p16/Ras or p53/p21 signaling pathways as signaling molecules, affecting cell cycle and accelerating cell senescence [81]. Increasingly more studies suggest that in addition to the accumulation of DNA damage, which is a regulatory factor for cell senescence, the decrease in genomic stability caused by chromatin structure changes has become an important regulatory factor in cell senescence. Epigenetic changes involve changes in DNA methylation patterns, histone posttranslational modifications, and chromatin remodeling. The Sirtuin family members of NAD-dependent protein deacetylases and ADP ribotransferases have been extensively studied as potential anti-aging factors [82]. One study showed that histone acetyltransferase inhibitors can also improve the premature aging phenotype of premature aging mice and prolong their life span [83].

The vascular endothelium is a single layer of cells adjacent to the vascular lumen and is a passive barrier layer between the blood vessel and the blood. Vascular endothelial cells can synthesize and secrete a variety of active substances that maintain blood flow; regulate vascular tension; regulate the production of proinflammatory molecules, proinflammatory immune response, and neovascularization; and maintain the integrity of vascular barrier and homeostasis of the internal environment [3].Vascular endothelial cell senescence is a complex biological process driven by gene regulation and environment, and its impact on the body is also complex. Senescent cells not only show a decrease in proliferation ability due to cell cycle arrest but also are accompanied by dysfunction, often manifesting as an aging-related secretory phenotype of pro-inflammatory and pro-oxidative stress, which is a potential risk factor for aging and aging-related diseases [84]. The proinflammatory phenotype of vascular endothelial cells is induced by various cytokines during senescence. Age-related inflammation is mainly characterized by increased C-reactive protein; pro-inflammatory factors such as TNF-a, IL-6, and intercellular cell adhesion molecule-1 (ICAM-1); and adhesion molecules in the blood circulation, and these inflammatory markers are positively correlated with the degree of arterial stiffness [85]. Previous studies have shown that there are pro-inflammatory changes in gene expression profiles of vascular endothelial cells in laboratory rodents and primates in the elderly. The pro-inflammatory microenvironment generated by vascular walls promotes vascular dysfunction and contributes to the pathogenesis of vascular diseases [84]. During endothelial cell aging and damage, the level of intracellular oxidative stress increases, the biological activity of reactive oxygen species (ROS) increases, and the bioavailability of NO decreases [86]. When NO secretion is insufficient or inactivated, it can lead to vascular endothelial dysfunction, which in turn promotes cardiovascular diseases such as hypertension, thrombosis, and atherosclerosis [87].

### 4.2. Dietary Anthocyanins Protect Cardiovascular System through Anti-Aging

Cardiovascular diseases are a serious threat to people′s health. At present, more and more studies have confirmed that vascular endothelial cell senescence may play a key role in endothelial dysfunction and aging-related vascular diseases [85,88]. Anthocyanins as berries and berry derivatives are particularly rich in polyphenols, with anti-endothelial cell senescence and cardiovascular disease protection having gradually been confirmed [5]. In recent years, a meta-analysis of randomized controlled trials (RCTs) was conducted to investigate the effects of anthocyanins from different sources on cardiovascular risk substitute markers such as hypertension, lipid profile, and endothelial dysfunction. The results showed that anthocyanins were beneficial to cardiovascular health [89,90,91]. Studies have shown that when the dynamic balance between oxidation and antioxidation is broken, excessive oxidative stress will accelerate the occurrence and development of aging and dysfunction of vascular endothelial cells [92]. Andrzej et al. isolated endothelial progenitor cells from the peripheral blood of young healthy volunteers and pretreated them with 1–25 µg/mL Aronia melanocarpa anthocyanin before angiotensin II treatment. The proliferation and telomerase activity of endothelial progenitor cells exposed to Aronia melanocarpa anthocyanin were significantly increased, and the percentage of aging cells and intracellular ROS formation were decreased compared with those of cells without Aronia melanocarpa anthocyanin pretreatment [93]. Li et al. showed that bilberry anthocyanin significantly increased the total antioxidant capacity, total superoxide dismutase activity, and catalase activity of aging mice, leading to decreased MDA, LDL-C, TC, TG, and GSP levels and a lower TC/HDL-C and LDL-C/HDL-C ratio. Phosphorylation of AMPK and FOXO3a, inhibition of mTOR phosphorylation, activation of autophagy induced by AMPK-mTOR signaling pathway, and improvement of oxidative-stress-induced senescence can be induced by intake of bilberry anthocyanins (20 mg/kg body weight/day) [94]. SIRT1 is a member of the silent information regulator 2 complex and is the main regulator of aging. The decrease in endogenous SIRT1 was found to damage acetylcholine-related, endothelial-dependent vasodilation and reduce the bioavailability of nitric oxide (NO) in aortic rings. Wang et al. have shown that anthocyanin-3-O-glucose can delay cell senescence by inhibiting inflammatory response, downregulating miR-204–5p, and upregulating SIRT1 [95]. Kun et al. improved the activity of SIRT1 deacetylase after pretreatment of endothelial cells with chlorogenic acid. In addition, chlorogenic acid (CGA) reversed the activity of SIRT1 and AMPK/PGC-1 induced by oxidized low-density lipoprotein and alleviated oxidative stress in endothelial cells induced by oxidized low-density lipoprotein [96]. Lee et al. [97] also proved that in vitro, anthocyanin-3-rutin (C-3-R) and anthocyanin-3-glucoside (C-3-G) could inhibit the senescence of human endothelial cells induced by d-gal; reduce the senescence-related β-galactosidase activity, p21, and p16INK4a; block the formation of reactive oxygen species (ROS) induced by d-gal and the activity of NADPH oxidase; and reverse the d-gal-mediated serine phosphorylation of endothelial nitric oxide synthase (eNOS) and the inhibition of SIRT1. In terms of the recovery of NO level in endothelial cells in vivo, after taking anthocyanin-rich mulberry extract for 8 weeks, aging rats were able to increase serum NO level, increase eNOS phosphorylation, increase SIRT1 expression, reduce nitrotyrosine in the aorta, and alleviate aging and oxidative stress of aortic endothelial cells. These results showed that anthocyanin increased the bioavailability of NO by regulating ROS formation and reducing eNOS uncoupling, thereby protecting endothelial cells from aging. Mi et al. found that anthocyanin can also inhibit the activation of NF-κB, downregulate the mRNA and protein expression of iNOS and COX-2, and protect oxidative stress-induced cell senescence [98]. Nrf2 is a NE-F2-coupling-array-binding factor in the globulin gene expression control region that is responsible for regulating cellular redox balance, protective antioxidants, and phase II detoxification in mammals. Nrf2 has been reported to play an important role in detoxification, oxidative stress, and inflammation. In addition, Nrf2 has been proven to play a role in slowing down the aging process. Hada et al. have shown that CGA can inhibit the degradation of Nrf2, enhance Nrf2 in the cytoplasm, and increase the expression of HO-1 mRNA and protein by regulating the Nrf2/HO-1 pathway, thereby inhibiting endothelial cell senescence [99].

## 5. Conclusions

Vascular endothelial cell senescence is a key factor leading to vascular homeostasis disorder and related diseases. Many mechanisms have focused on the inhibitory effect of anthocyanin and its metabolites on endothelial cell senescence. Anthocyanins and their metabolites regulate the clearance of endothelial cell senescence by controlling the activity of cell signal proteins and transcription factors and regulating the expression of genes and miRNAs. However, the potential mechanism for dietary anthocyanin to play an active role in maintaining cardiovascular health is complex and not fully determined. We also need to design more comprehensive and holistic experiments to study the effects of anthocyanin and its metabolites on gene, protein, and miRNA expression; provide reliable evidence to determine the exact role of dietary anthocyanin in cardiac protection; and explain its potential molecular mechanism.

## Figures and Tables

**Figure 1 nutrients-14-02836-f001:**
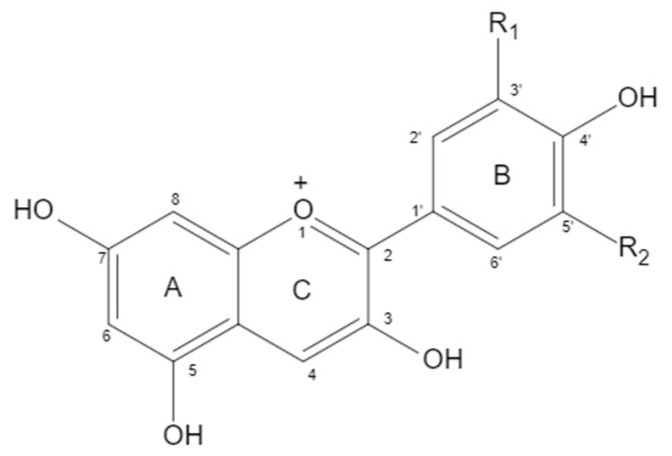
Chemical structure of anthocyanins.

**Figure 3 nutrients-14-02836-f003:**
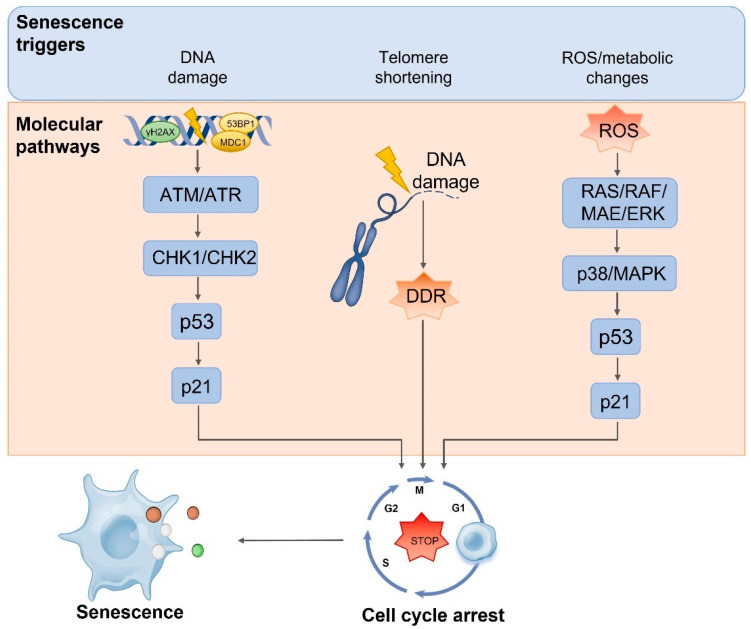
The molecular pathways involved in senescence cell cycle arrest. DNA damage activates a signaling cascade, defined as DNA damage response, which is characterized by phosphorylation of histone H2AX, 53BP1 and MDC1, ATM and ATR, and downstream kinases CHK2 and CHK1. The signal finally concentrates on the activation of p53, which induces p21, thereby causing cell cycle arrest. In the process of DNA replication, telomeres shorten to the limit, leading to DNA double strand breaks, triggering DDR, causing cell cycle arrest. The excessive production of reactive oxygen species (ROS) will also challenge the integrity and stability of DNA, or directly regulate the p53/p21 signaling pathway as a signal molecule, affect the cell cycle, and accelerate cell aging. Anthocyanins have a significant antioxidant capacity that can effectively remove ROS, repair damaged cells, and relieve cell cycle arrest. Abbreviations: ROS, reactive oxygen species; γH2AX, phosphorylated histone H2AX; ATM, ataxia–telangiectasia mutated; ATR, ATM and Rad3-related homologue; DDR, DNA damage response; MAPK, mitogen-activated protein kinase.

**Table 1 nutrients-14-02836-t001:** Chemical structure of anthocyanins.

Anthocyanin	Proportion	Substituents	
		R_1_	R_2_
Cyanidin	50%	OH	H
Delphindin	12%	OH	OH
Pelargonidin	12%	H	H
Peonidin	12%	OCH_3_	H
Petunidin	7%	OCH_3_	OH
Malvidin	7%	OCH_3_	OCH_3_

**Table 2 nutrients-14-02836-t002:** Physiological activities of anthocyanins.

Effects		Source	Mechanisms	Ref.
Anti-cancer	colon cancer	Purple grape anthocyanins	Inhibited IκBα phosphorylation Prevented tumor necrosis factor α-induced NF-κB activation	[13]
	colon cancer	Cyanidin-3-O-rutinoside	Reduced the motility and the metastasis	[14]
	colon cancer	Purple and red maize anthocyanins	Enhanced BAX, Bcl-2, cytochrome C, and TRAILR2/D5 Inhibited Tie-2, ANGPT2, and PLG	[15]
	breast cancer	Black sweet cherry anthocyanins	Downregulated Sp1, Sp4, and VCAM-1	[8]
	melanoma cancer	Hibiscus calyx anthocyanin	Triggered PI3K/Akt and Ras/MAPK signaling pathways Downregulated VEGF and MMP-2/-9	[16]
		Purple sweet potato anthocyanin	Acted on cell cycle regulators (such as p53, p21, p27, Cyclin D1, and Cyclin A)	[17]
Anti-inflammatory			Nrf2-ARE signal modulation	[20]
			Inhibited C/EBP, AP-1, and NF-κB Inhibited COX-2 Reduced IL-1β, IL-6, IL-8, and TNF-α Enhanced *PPAR-γ* gene	[21] [24]
		Cyanidin-3-glucoside	Regulated NF-κB and MAPK activity	[22]
		Geranium pigment-3-O-glucoside in strawberry	Inhibited the activation of IkB-α Reduced the phosphorylation of JNK-MAPK	[26]
		*Hibiscus* anthocyanin	Inhibited the secretion of nitric oxide and prostaglandin E2 Reduced MyD88 growth and IRAK4 phosphorylation Inhibited NF-κB activity	[27]
Anti-oxidation		Purple corn anthocyanin	Had DPPH radical scavenging activity	[30]
		Cyanidins in radish buds	Inhibited the automatic oxidation of linoleic acid Scavenged hydrogen peroxide free radicals	[31]
		*Mahonia aquifolium* anthocyanin	Had DPPH and ABTS radical scavenging ability Had FRAP reduction ability	[32]
		Black rice anthocyanin extract (cyanidin-3-O-glucoside)	Improved the activities of superoxide dismutase and catalase Reduced the content of malondialdehyde and the activity of monoamine oxidase	[33]
		Blueberry anthocyanins	Decreased the levels of ROS and XO-1 Increased SOD and HO-1	[34]
Protective effect on liver		Cyanidin-3-O-glucoside	Prevented fibrosis Reduced liver oxidative stress Reduced liver cell apoptosis Inhibited liver inflammatory response	[36]
		Riceberry bran anthocyanin	Inhibited intracellular oxidative stress and the activation of NF-κB factor Reduced liver cell inflammation and apoptosis	[37]
		Purple sweet potato anthocyanin	Activated adenosine-monophosphate-activated protein kinase (AMPK) signaling pathway Inhibited the production of reactive oxygen species Inhibited the accumulation of liver fat	[38]
		Purple sweet potato anthocyanin	Had obvious protective effect on the release of alanine aminotransferase (ALT)	[39]
Lowering blood glucose		Purple corn anthocyanins	Protected pancreatic β cells from high-glucose-induced oxidative stress Improved insulin secretion ability of β cells	[52]
		Mulberry anthocyanin	Increased AMPK phosphorylation Inhibited gluconeogenesis and stimulated glycogen synthesis	[41]
Anti-aging		Purple sweet potato anthocyanin	Reduced the serum MDA level Improved the activities of SOD and GSH-PX Delayed aging by improving antioxidant activity	[45]
		Cy-3-glu Pg-3-glu	Inhibited the galactosidase	[46]
		Ribes meyeri anthocyanins	Promoted the proliferation of neural stem cells Improved cell senescence phenotype Reduce ROS Reduced senescence-associated *P16Ink4a* gene expression levels Increased DNA synthesis Prolonged telomeres	[47]
			Maintained the stability of redox system Reduced the levels of IL-1, IL-6, and TNF-α Decreased in the expression levels of sensors, media, and effectors in the DNA damage signaling pathway Slowed down aging by inhibiting DNA damage	[48]

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
