# Peer review of "The Potential Roles of Dietary Anthocyanins in Inhibiting Vascular Endothelial Cell Senescence and Preventing Cardiovascular Diseases"

_nutrients, 2022, doi:10.3390/nu14142836_

Round 1

Reviewer 1 Report

In this review article, the authors have summarized information regarding bioactivity of anthocyanins and their gut microbiota-mediated metabolism, as well as their inhibitory and preventing effects on vascular endothelial cell senescence and dysfunction related to cardiovascular disease. Overall, the current manuscript is well-written logically, and should be of potential interest to the readership of the journal. However, the authors could significantly strengthen their manuscript by addressing the following major concerns. Firstly, the current manuscript is quite descriptive and lacks supporting illustration and/or a kind of graphical scheme that help readers understand concepts of this manuscript regarding mechanism of vascular endothelial cell senescence as well as absorption and metabolism of anthocyanins. Secondly, the reviewer is seriously concerned whether all of the references are properly cited, and whether the current description is fully supported by rigorous study without overestimation and/or overstatement. The authors should check again to confirm all of the citations are adequate to assure the authors’ explanation, since the reviewer found some parts lacking appropriate evidences and citations that may be somewhat overstatements. Thirdly, many parts including Table 1 were found to be simply lists of previous studies. The authors should organize and include discussion with respect to what is known or not as limitation of studies.

- One of the major topics in this review article, endothelial cell senescence/aging and their implication for cardiovascular diseases, is separately described in chapter 2 and 5 and some repetitive sentences exist. To make it consistent, the reviewer recommend that chapter 2 is integrated in the first part of section 5.

-   Line 47-49: Appropriate references should be included.

-  To explain characteristics and mechanism of endothelial cell senescence described in line 58-105, preparation of illustration or schematic diagram would be valuable.

-  Line 88-105: The reviewer understand that vascular endothelial sell senescence consists of certainly complicated biological process as the authors documented. However, it is important to introduce its overview with appropriate references such as “Circ Res 2018;123:825-848”.

-  Line 101: As the study in Ref 12 was investigated using vascular smooth muscle cells, other proper references should be cited in line 98-101 regarding endothelial dysfunction associated with aging.

-   The chemical structure of anthocyanins as well as the table shown in Fig 1 is of poor quality. Resolution should be improved.

-   Line 244-251: It is difficult to confirm and follow whether appropriate analyses were performed to investigate anthocyanin-mediated decrease in sugar content in the serum and glucose consumption on the basis of previous studies. Since ref 49 is a review article, it should be replaced to several proper regular articles. In addition, ref 50 is not appropriate to support anthocyanin-induced reduction of blood glucose since in vivo study was not performed in the reference.

-   Line 252-261 consists of less citations and less direct evidence for anti-aging effects, therefore “anti-aging” remain poorly defined. Additional detailed description would be necessary.

-  Table 1 is the mere repetitive of the main text, just looks busy and complicated. That should be organized and simplified with categorized keywords such as effects, mechanism, pathway, mediators, in vitro or in vivo (human or animal experiments), meta analysis and so on.

-  Some references should be cited in line 344-349 to support the authors’ description.

- Process of anthocyanin metabolism including degradation (deglycosylation), absorption occurred in small intestine, interaction with gut microbiota should be illustrated as a kind of a schematic view to help readers understand the description.  

-    Line 387: “between the blood and the blood” should be corrected.

-   Line 390: Ref 78 seems to be old article. Other review article such as “Circ Res 2018;123:825-848” should be also cited.

-  Line 418: The abbreviation “CGA” should be indicated just after “chlorogenic acid”. Accordingly, “chlorogenic acid” in line 439 should be abbreviated.

Author Response

请参阅附件。

Reviewer 2 Report

This paper does not stand out from other literature in the field. Scientific review articles must be critical analyses of available information about a specific topic. This article lacks the latest references to demonstrate recent findings. Although there are some data representations through tables but lacking presentation through figures. Mechanisms that are discussed here should be in figure form to make it easy for the readers.  

Author Response

请参阅附件。

Round 2

Reviewer 1 Report

The authors have revised this article and the manuscript has been improved with additional graphical information etc, but it still riddled with some serious issues.  

-   Brief overview of Fig. 3 needs to be described as figure legend of Fig. 3, and all abbreviations should be explained. Moreover, molecular and/or potential targets of anthocyanin for prevention of aging and cell senescence need to be indicated in Fig. 3 to summarize key concepts of this review article.

-  References, especially added in the process of revision, are not balanced, with too much emphasis on studies in east Asia.

-    Gene names should be described in italic font.

-    References should be in order.

Reviewer 2 Report

Authors made efforts to improve the quality of article.
